# Review on the Application of Machine Vision in Defrosting and Decondensation on the Surface of Heat Exchanger

**Bin Yang [1,*], Xin Zhu [1], Minzhang Liu [1] and Zhihan Lv [2]**

1   School of Energy and Security Engineering, Tianjin Chengjiang University, Tianjin 300384, China
2   College of Art, Uppsala University, s-75105 Uppsala, Sweden
*   Correspondence: binyang@tcu.edu.cn

**Abstract:** Under low outdoor temperature and high humidity, frost easily forms on the Heat Exchanger (Exchanger) surface on the outdoor side. The formation and growth of this frost layer will seriously impact the Exchanger's heat extraction process and the system's energy efficiency, triggering malfunction in the compressor. To this end, this work first analyzes the formation and growth mechanism of Exchanger surface frosting and condensation. It then summarizes the current research status of Machine Vision (MV) technology in defrosting and decondensation. Further, it previews the follow-up research direction. The experimental findings show that MV technology can automatically observe frost and dew, guaranteeing a real-time understanding of the frost layer. Directly obtaining the frost and dew information from the image can significantly save human resources and improve efficiency.

**Keywords:** indirect evaporative cooling; near-zero energy consumption buildings; measured analysis; cooling efficiency; temperature drop

## 1. Introduction

Social progression and upgraded comfort in living are demanding more user-friendly Air Conditioning (AC) products to maintain a relatively stable indoor environment. Inevitably, these AC systems consume large amounts of energy. In China, heating and cooling networks have become the leading energy consumption source in buildings. Currently, the domestic energy supply is mainly generated by oil and coal, with a low Resource Utilization Rate (RUR) due to technological backwardness. As a result, Partial Oxidation aggravates environmental pollution caused by the carbon-intensive economy, harming the working and living environment. In particular, the Air Source Heat Pump (ASHP) features low cost, convenient installation and use, high efficiency and zero pollution. Therefore, it has broad applications in the "coal to electricity" energy reform. Meanwhile, frost will be formed on the outside surface of the Heat Exchanger (Exchanger) under low temperature and high humidity. The continuous growth and accumulation of the frost layer seriously impact the Exchanger's heat extraction, thereby reducing the system's operation efficiency or even triggering compressor crashes.

Many scholars have carried out research into and exploration of on the defrosting problem on the surface of the Exchanger. Zhou and Cheng (2001) [1] confirmed that the heat source of hot gas defrosting came from the inside of the evaporating coil; the effect of temperature fluctuations in the cold storage was small; the lower the temperature of the storage, the more prominent the energy saving effect of hot air defrosting. Abdulla et al. (2021) [2] studied the growth mechanism of frost layer on the shutter fin of microchannel Exchanger through the image segmentation method and frost layer density distribution and summarized the three stages of the frost process and the characteristics of frost layer growth. In addition, they studied the influence of the residual condensate water on the surface of the Exchanger in the frosting process and quantitatively gave the relationship between

the residual water amount and the frosting time, the maximum heat transfer capacity and the wind resistance. The authors also explored the drainage frosting performance of microchannel Exchangers in different locations and efficiently improved the drainage capacity of blind fin Exchangers through vertical flat tube placement. In their study, the stability of the circulating frosting process was guaranteed and the effective operation time was increased by 32%. Lee et al. (2021) [3] established the air heat transfer and flow model outside the evaporator tube, the evaporator tube wall model, the heat transfer and flow model of the working medium inside the evaporator tube and the frost growth model outside the evaporator, via the mass and energy conservation equation. In addition, they established the dynamic distribution parameter simulation model of frosting and heat transfer in a finned tube evaporator, providing a critical support for the development of defrosting research in Exchangers.

Thermal defrosting and mechanical defrosting are two broadly used defrosting methods at present. Thermal defrosting refers to melting the frost layer on the surface of the heat exchanger by heating, including reverse circulation defrosting, hot air bypass defrosting, hot water spraying defrosting, electric heating defrosting and compressor shutdown defrosting, etc. Mechanical defrosting relies on a mechanical external force to remove the frost layer on the surface of copper tubes and fins, including ultrasonic crushing, high-pressure air blowing and mechanical scraping. Solving the problem of frost on the surface of the Exchanger can effectively improve the comprehensive energy efficiency of the heat pump, enhance the comprehensive utilization efficiency of energy in China and achieve the goal of energy saving and emission reduction. This work explains the condensation and frosting mechanism on the surface of the Exchanger based on the principle of thermal defrosting technology and summarizes and analyzes the various stages of frosting on the surface of the Exchanger. The defrost and frost suppression phenomena which are common at present are emphasized to provide a developmental direction for intelligent defrosting and frost suppression technology on the Exchanger surface by a thermal defrosting method.

## 2. Formation Mechanism of Frost and Dew; Application Status of Machine Vision in Defrosting and Dew Decondensation

*2.1. Formation Mechanism of Dew and Frost*

2.1.1. Surface Frosting Principle of the Exchanger

Frost layer formation has been a significant concerned for some time. Recently, Machine Vision (MV) technology and visualization equipment have reshaped people's understanding of the frost layer formation process. According to the initial frosting stage, the frost formation can be divided into condensation frost and hoarfrost. Water vapors in the air first condense into droplets and then freeze into frost crystals, i.e., condensation frost. In comparison, in hoarfrost formation, water vapors directly appear from a gaseous state as frost crystals and grow into a frost layer [4]. The Gibbs Free Energy (GFE) barrier must be overcome in both condensation frost and hoarfrost formations. However, the former's GFE barrier is smaller than the latter. Therefore, condensation frosts are much more common, while hoarfrost only occurs on unique surfaces or individual extreme environments [5]. Accordingly, researchers' attentions focus on condensation frost.

Hayashi et al. (1977) phased frost layer formation into the initial frost-layer growth and the complete frost-layer growth stages. In the initial stage, the water vapor in the air condenses as ice crystals on the cold surface. Over time, the ice crystal area surface expands and then shapes the frost core. Then, new ice crystals spread to different directions around the frost core. The initial condensation process is vital for the subsequent frost layer growth, which is affected mainly by the characteristics of the cold surface [6]. An accurate description of the internal structure of the frost layer and the accurate determination of thermophysical properties have always been problematic in researching frost formation mechanisms. Zhang (2005) [7] aimed to reveal the frost layer growth structure in the early frosting stage and developed a set of experimental frosting devices to observe the image of the frost layer growth process. By controlling frosting conditions, the research

provided an experimental basis for analyzing the characteristics of the frosting process. Moreover, the device also solved the problem of unclear shooting and many other minor problems when obtaining experimental frosting images in China. The proposed devices could provide clear images with large magnification. Hou (2006) [8] systematically studied the frost layer growth process and its influencing factors. Microscopic photography was used to obtain frost layer growth images in different periods and working conditions. Then, the digital IP method analyzed the frost growth images in different periods. The frost structure was studied based on fractal theory. Additionally, the nucleation of the water vapor phase and the heat and mass transfer in the frost layer was discussed in order to implement the frost layer growth model. The proposal laid a foundation for inhibiting the growth of the frost layer and the selection of defrosting time. Hou et al. (2007) [9] observed the morphology of ice crystals at the initial stage of frost growth through the self-developed image amplification and acquisition system at a microscopic level. The images of ice crystals under different growth conditions were obtained. Then, the original image was transformed into a binary image by digital IP to perform fractal analysis by the box-counting dimension method. Liu et al. (2009) [10] proposed a two-dimensional (plane) frost crystal growth model based on the Diffusion Limited Aggregation (DLA) growth model of fractal theory. The simulation results implied that the images of frost formation and the growth process simulated by the fractal theory model were in good agreement with the experimental images. Liu et al. (2012) [11] used the digital IP function of MATLAB to integrate the incomplete boundary grid with binary image meshing, optimized the box-counting dimension of the binary image and improved the accuracy and stability of calculation results.

The frost crystals show needle and dendritic growth patterns at the initial frost formation stage on the cold surface. Gradually, they become flat shapes. The frost layer growth stage follows after the frosting crystal covers the whole of the cold surface. Then, the frost layer thickens. Its surface temperature can become high enough to melt the frost crystal and decrease it periodically. Due to the growth mode of the frost crystal, its fractal dimension shows a multi-level increasing trend with time. Liu et al. (2012) [12] discussed the physical significance of fractal dimension and volume fraction in describing the characteristics of the frost layer. They pointed out that fractal parameters should be introduced into frost process modeling. Liu (2012) [13] conducted cold surface frosting experiments under different experimental conditions. Then, they measured the changes in frost layer thickness with different frosting times or frosting conditions, as well as frost layer surface temperature, frosting rate and frost crystal morphology. Meanwhile, the fractal characteristics of frost crystal growth were studied and the frosting model combining fractal theory and phase transition dynamics theory was proposed. Chen et al. (2012) [14] numerically simulated the frost layer growth process on the surface of the finned tube gasifier based on the DLA model of fractal theory. They experimentally observed the frost layer growth morphology and obtained frost layer growth images at different times. Specifically, the experiment calculated the fractal dimension and fractal porosity of the pore area distribution of the frost layer profile on the surface of the gasifier. Comparison of the simulated image and the experimental image found that they were in good agreement, verifying the rationality of the numerical simulation. On this basis, the fractal model of frost layer heat conduction was established and the expression of frost layer thermal conductivity was given by the thermal resistance method. Yao et al. (2013) [15] used the designed vertical plate frosting testbed to conduct experimental research on frosting on the cryogenic surface under natural convection conditions. They obtained the variation law of the cryogenic surface temperature, frost layer thickness and heat flux through the cryogenic surface over time in the early frosting stage. At the same time, the growth form of the frost layer was observed experimentally to obtain the frost layer growth images at different times. Finally, the heat flux passing through the cryogenic surface during frost formation was calculated based on the fractal analysis of frost layer growth and the fractal theory's effective thermal conductivity of the actual frost layer. Yao (2013) [16] set up an experimental platform

for frost layer growth on the cryogenic surface. The changes in frost layer growth with frosting time were experimentally studied, as well as frost layer thickness, cryogenic surface temperature, heat flux change passing through the cryogenic surface and wet airflow state. Consequently, frost layer growth images were obtained in different periods. In particular, fractal theory was introduced to quantitatively analyze the effect of frost growth on the heat transfer performance of the frost layer under different working conditions. By summarizing the development and research status of defrosting of ASHP in and outside China, Hang (2017) [17] devised an ASHP-oriented defrosting control method based on IP technology and tested the defrosting control method experimentally.

Boreyko et al. (2016) observed the frosting process of hydrophobic surfaces and divided the process into droplet growth, growth, freezing, frost crystal formation and further growth. Consequently, they claimed that the droplet expansion rate on the hydrophobic surface was lower than that of the ordinary surface. Thus, a hydrophobic surface could inhibit the growth of the frost layer [18]. The frost layer density increases in the growth stage and thickness does not change much. Then, the frost layer enters the full growth stage with continuous growth. The heat released by water vapor condensation will increase the frost layer surface temperature. Then, the frost layer melts partially into water droplets that penetrate the bottom frost layer through the frost layer gap. Further, these droplets condense into ice again on the colder surface. At this stage, the ice thickness, frost density and thickness of the cold surface will continue to increase.

### 2.1.2. Growth Mechanism of Frosting on Exchanger Surface

Liquid Film Rupture Hypothesis and Fixed Center Nucleation Hypothesis are two research viewpoints on condensation. The former theory believes that water vapors in the air will first form a water film on the cold wall through liquefaction. Then, many tiny water droplets will be generated once the water film swells sufficiently and breaks. The second theory holds that the water vapors will first form a liquid core on the cold surface through condensation and droplets will grow around the core. So far, the Fixed Center Nucleation Hypothesis is supported by most scholars.

Condensation is the process of saturated water vapors in the air condensing and precipitating into water droplets [19]. Mainly, two factors contribute to the condensation phenomenon. (1) When the partial pressure of water vapor in the air is fixed, the temperature of unsaturated wet air will drop after cooling. Then, the saturated partial pressure of water vapor will drop until the temperature reaches the dew point. At this time, the water vapor will liquefy. (2) Under a fixed indoor temperature, if the partial pressure of water vapor rises until the saturated moisture content is reached, the water vapor in the wet air will liquefy.

According to the condensation location, the Fixed Center Nucleation Hypothesis divides condensation into surface condensation and homogeneous nucleation condensation [20]. The former is formed on the surface of the cold wall that directly contacts the wet air. The latter is not the result of direct contact with the solid surface. Water droplets are formed in the air and the whole process takes place in the wet air space.

### 2.2. Researching Defrosting and Decondensation on Exchanger Surface

### 2.2.1. Research Status of Surface Defrosting Methods for Heat Exchangers

In recent years, researchers have suggested a series of methods and measures for defrosting Exchanger surfaces to improve the Exchanger heating performance in low-temperature environments. Frost Suppressing can improve the Exchanger's overall operational efficiency. According to the implementation objects, Exchanger Frost Suppression methods are divided into internal and external methods. Internal Frost Suppression changes the internal conditions, such as refrigerant flow distribution, surface modification treatment and Exchanger structure design. In contrast, external Frost Suppression adjusts the external conditions, such as the ElectroMagnetic field and air parameters at the Exchanger inlet. The following will summarize and analyze the standard Frost Suppression methods.

(1)    Changing the inlet conditions of the Exchanger

There are several Exchanger frosting conditions. First, the dew point is higher than the Exchanger surface temperature so that the water vapor in the air can condense. Second, the Exchanger surface temperature is lower than 0 °C. Adjusting the Exchanger inlet conditions can effectively inhibit frosting on the Exchanger surface. Zhu et al. (2015) mapped the Exchanger surface frosting according to ambient air condition influence into three operation zones: condensation zone, frost-free zone and frosting zone [21]. Wang et al. (2017) built a solid desiccant-based heat pump system. The search utilized the dehumidification effect of solid desiccant. It regenerated the desiccant using storage heats in the phase-change accumulator. As a result, the system could generally operate without frost, even with a decreased dehumidifier performance [22]. The factors affecting the Exchanger frosting are detailed in Figure 1:

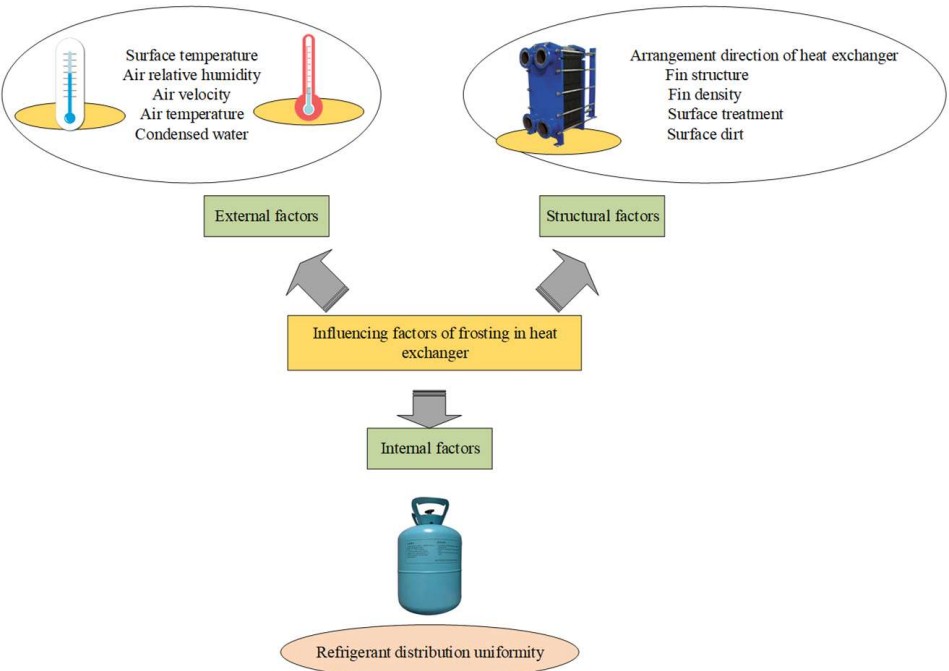

**Figure 1.** Factors affecting Microtube Heat Exchanger (MTHE) frosting.

(2)    Optimizing the structure and design of the Exchanger

Adjusting the fin spacing and structure of the Exchanger can improve its performance to suppress frosting.

Peng et al. (2021) tried to suppress Exchanger frost and improve its operation time and performance by optimizing the finned tube parameters [23]. Using the optimization scheme significantly enhanced the Exchanger's mean running time and efficiency. Park et al. (2016) examined the impact of louver fin size on Exchanger surface frosting. The research found that an Exchanger with louver fins could suppress the frost layer accumulation between fins and support heat exchange performance [24].

Different fin-type designs will also affect the Frost Suppression effect of the Exchanger. Liu et al. (2018) simulated and compared the frosting and defrosting processes of three Exchanger types in domestic ASHP. The experimental results implied that, regarding the outdoor Exchanger defrosting effect, the flat fin Exchanger outperformed the Louver Fin Exchanger and Corrugated Plate Heat Exchanger (CPHE) [25]. Wang et al. (2018) studied the operation effects of three Exchangers: Round Tube Corrugated Fin (RTCF), Parallel Flow Parallel Fin (PF2) and Parallel Flow Serpentine Fin (PFSF) under frosting, humidity and drying, respectively. The research outcome was that the running time of the PF2 Exchanger

was the shortest and that of the RTCF Exchanger was the longest [26]. Although optimizing the Exchanger structure can achieve a defrosting effect, the effectiveness is still limited.

(3)    Surface Modification and Frost Suppression of Exchanger

Due to the hydrophobic property of the surface material and its rough structure, the adhesion force of the superhydrophobic surface is far less than that of the ordinary surface. Therefore, the residence time of the frost layer or icing on the superhydrophobic surface will not be for too long. Inspired by the pitcher (an insectivorous plant), Dong and Li (2020) proposed a new method for preparing smooth liquid filled with a porous surface. They filled the micro nanostructure of the solid surface with immiscible low surface energy lubricant to form a smooth solution layer on the surface [27]. Solution lubricant and oil lubricant were usually selected for Exchanger surface defrosting. Du et al. (2021) investigated the defrosting water retention on surfaces with different wettability. Theoretically, they analyzed the retention mechanism of defrosting water. The experimental results implied that the contact angle lag would significantly impact the frost retention water on the Exchanger's surface. Compared with the ordinary surface, the self-adhesion of the superhydrophobic surface was lower and the surface retention water was reduced by 79.82% [28]. Chu et al. (2018) surveyed the defrosting mechanism in vertical superhydrophobic surfaces. The results corroborated that, when the frost layer thickness reached a threshold, the frost layer would fall off. However, the frost layer would fall off when the thickness was less than the threshold [29]. Wang et al. (2018) at the Beijing University of Technology explored the impact of different defrosting cycles on the operating performance of ASHP. They sought the best defrosting cycle under different frosting conditions. Five different defrosting cycles were used to measure and research the unit under four different frosting conditions in Beijing. The measurement was carried out under an ambient temperature of $-2.1{\sim}5.6\,^\circ C$, relative humidity 48%~88% and return water temperature 40 °C. The results indicated that an optimal defrosting cycle existed in the unit under the same frosting condition and would change with different frosting conditions. The nominal defrosting loss coefficient is 24%~36% when the optimal defrosting cycle is adopted for defrosting control. Meanwhile, the nominal effective heating coefficient could reach 0.69–0.78 and the average performance coefficient remained at 2.06–2.30. The unit operated significantly better than other defrosting control cycles [30].

2.2.2. Effect of Surface Condensation on Exchanger

In the 1980s, American scholars first put forward the concept of MTHE. The hydraulic diameter of the MTHE tube ranges from tens to hundreds of microns. The components include refrigerant inlet and outlet, fins, header and flat tube. The flat tube connects the two headers and contains multiple channels inside. MTHE is widely used in AC systems because of its high heat exchange efficiency, low refrigerant charge and small volume. They are mostly used as condensers in commercial refrigeration and AC. However, during summer, condensation often occurs when MTHE acts as an AC dehumidification evaporator. In addition, MTHE has a fin structure that cannot effectively eliminate the surface condensation. This often leads to increased thermal and flow resistance on the air-side and decreased heat exchange effect.

Kang et al. (2021) researched the e-vehicle AC system to explore the influence of inlet air temperature on the heat transfer characteristics of the MTHE evaporator. The experiment revealed that the evaporator loss was smallest and the largest under an inlet air temperature of 42 °C and 27 °C [31]. Reda et al. (2021) numerically simulated the air-side flow and heat transfer characteristics of MTHE. They observed that the heat transfer efficiency on the MTHE air-side would be enhanced by increasing the angle of louvers. Simultaneously, this reduced the spacing and increased the wind speed on the MTHE air-side [32]. Ali and Muneer (2021) investigated the impact of airflow rate on the air pressure drop coefficient on the front side of the condenser through the experiment. As a result, the air-side wind resistance, heat transfer coefficient and wind resistance would increase with frontal air velocity. The Heat Exchange Coefficient (HEC) would decrease

with head-on air velocity [33]. Shamirzaev (2018) scrutinized the relationship between air-side pressure drop of aluminum MTHE and HEC, heat exchanger inclination and inlet air humidity. The numerical results claimed that the air-side pressure drop would increase with the inlet humidity. However, when the inlet air humidity was lower than 45%, the air-side pressure drop was not affected by the air humidity [34].

## 3. Application Status of MV in Defrosting and Dew Decondensation

### 3.1. Application of MV Technology in Exchanger System

3.1.1. Intelligent Defrosting Technology

The Exchange system can realize different requirements by controlling the electronic expansion valve and compressor speed. With the development of Artificial Intelligence (AI), Internet of Things (IoT), Cloud Computing (CC) and MV, scholars have put forward a series of control strategies for Exchanger defrosting problems. The existing Exchanger surface defrosting problems fall into three categories: direct defrosting, indirect defrosting and intelligent defrosting. Li et al. (2017) of Tsinghua University used four models: multiple linear regression, Neural Network (NN), least-squares Support Vector Machine (SVM) and adaptive neuro-fuzzy inference system to estimate the frost layer thickness of the vertical low-temperature surface. In order to evaluate the different models' performance, different graphics and statistical error analyses were used to compare the results with the measured data in the laboratory. The results proved that the adaptive neuro-fuzzy inference system was superior to other models [35]. Li et al. (2021) researched applying image recognition to the ASHP-oriented frost detection. The results revealed that the improved frost detection method could better consider the influence of illumination change on image recognition accuracy. This ensured frost detection accuracy for ASHP operating under frosting conditions [36].

The direct defrosting process mainly involves Photoelectric Coupling (PEC), Micrometer Thickness Measurement (MTM), probe-based MTM, Laser-based thickness measurement and Microscopic Imaging (MI) technologies. Specifically, PEC uses photoelectric technology to judge the frosting condition on the Exchanger surface. Probe-based MTM observes the micromorphology of the frost layer using a probe. By comparison, MTM and laser-based thickness measurement technologies employ micrometer-level meters and laser sensors, respectively, to measure the thickness of the frost layer. MI technology first utilizes the micro camera to capture the frost image and then outputs the frost layer thickness through IP technologies.

Below, we give some examples of indirect defrosting control methods: refrigerant superheat-based defrosting, maximum Mean Heating Capacity (MHC)-based defrosting, Air Pressure Difference (APD)-based, temperature defrosting and timing defrosting. The refrigerant superheat-based approach measures the refrigerant outlet superheat to determine whether to defrost the Exchanger. The maximum MHC-based method defrosts when the MHC of the heat pump medium reaches the maximum. The APD-based defrosting method counts the APD on the working medium side to determine whether to defrost. Lastly, temperature defrosting estimates the Exchanger surface temperature to judge whether to defrost. The time defrosting rule performs regular defrosting by setting the defrosting cycle. Additionally, Wang et al. (2017) from the Beijing University of Technology introduced two new defrosting control methods: optical electric conversion based on direct measurement and temperature humidity time based on indirect measurement. The accuracy and efficiency of the two new defrosting control methods were verified through field tests [22].

Recently, intelligent defrosting control methods have caught researchers' eyes, including NN control and Adaptive Fuzzy Control (AFC) methods. For example, according to the experimental data, the NN-based control method constructs a nonlinear relationship between the frosting variables. The AFC method judges the effectiveness of the control rules. It adaptively adjusts the frosting control after fuzzy reasoning and fuzzy processing of the input variables. Li and Deng (2017) modeled the Variable Speed Direct Expansion (VSDX) AC system using a steady-state NN. Simultaneously, they ensured the relative

system output apparent and latent cooling capacity did not change significantly under different evaporator inlet air conditions. The proposed model predicted the system output under different compressor and fan speed combinations. Using the relative value of the output apparent and latent cooling capacity could eliminate the influence of the indoor air state on the system output. The steady-state experimental data were obtained to train, detect and verify the accuracy of the steady-state NN model in predicting the operating characteristics of the VSDX system. This experiment verified the applicability of the proposed hypothesis and the proposed model at the point of a non-training state. The steady-state NN model's maximum training, detection and verification errors were less than 5%. The average error was less than 3%. The findings indicated that the steady-state NN model could accurately predict the operating characteristics of the VSDX system at the training state point and the non-training state point [37]. To avoid the adverse defrosting phenomenon, Zheng et al. (2019) proposed a new defrosting control method based on IP technology: the temperature-humidity image method. As a result, image preprocessing, gray transformation and multi-threshold segmentation were used to quantify the frosting visual features. The surface was divided into a non-frosting area, a medium frosting area and a severe frosting area [38].

When the ambient air humidity and temperature reach a threshold, the Exchanger surface frosting becomes the main factor affecting the operational efficiency of ASHP. The traditional ASHP defrosting control methods cannot meet the urgent needs of new application scenarios. Against the heat supply and system performance degradation caused by Exchanger surface frosting, Ye et al. (2021) conducted an in-depth study on the frosting and defrosting process of ASHP units under different operating environments. Consequently, an intelligent defrosting control logic was proposed based on MV technology, snow prevention control, large flow defrosting control methods and anti-icing and defrosting control methods. Then, the experiment was simulated to analyze the ASHP units. The experimental results show that optimizing control and intelligently selecting various defrosting control methods could meet the defrosting requirements under different environmental conditions. Meanwhile, the defrosting time could be significantly shortened and a practical solution was provided for the performance improvement, popularization and application of ASHP [39]. Miao et al. (2022) tried to accurately control the defrosting time, avoid poor defrosting and improve the defrosting process by proposing a new temperature and humidity image defrosting control strategy. Consequently, the proposed strategy could accurately and effectively perform defrosting operations based on IP technology and fractal analysis. Compared with the traditional methods, the proposed temperature and humidity image defrosting control strategy effectively avoided delayed defrosting and unnecessary defrosting. Meanwhile, the defrosting energy consumption was reduced by 10.6% and 22.3%, respectively [40].

MV technology exercises machines to replace human eyes for observation and judgment. The composition of the MV system is shown in Figure 2:

MV techniques provide a new idea for image-based frost weather recognition. However, the current research mainly focuses on road safety and traffic scenes, with limited recognition of scenes and categories. There is a lack of representative, large-scale and rich-type data sets. The algorithm's accuracy needs to be improved. Therefore, citing the current MV literature, expanding the recognition scene and establishing a wider variety of data sets are the key to the development of MV technology. The MV-based recognition flow is demonstrated in Figure 3:

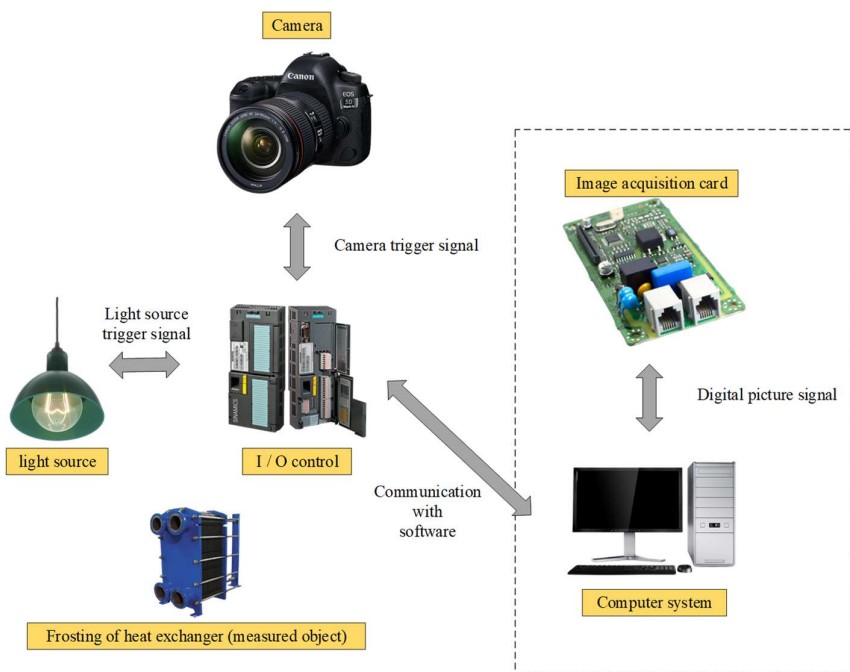

**Figure 2.** Composition of MV system.

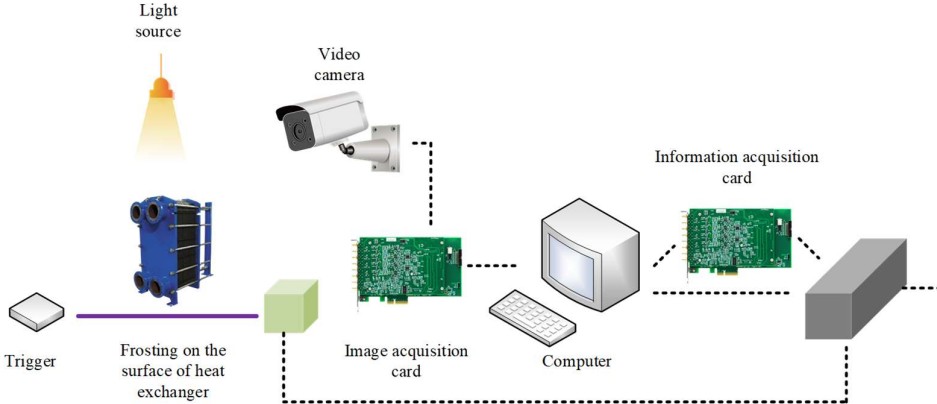

**Figure 3.** Flow of MV recognition.

Apparently, the MV-based recognition system first captures the target image and then converts it digitally using the camera. Afterward, the digital image is collected through the image acquisition card. Some helpful information can be extracted using IP technology and the computer. Finally, the processing result is exported [41]. MV technology has effectively improved the accuracy and precision of target detection.

### 3.1.2. ASHP

ASHP is an energy-saving device using high-level energy to transfer heat from low-level heat source air to high-level heat source. It is a form of heat pump, similar to a traditional pump, which can convert low-level heat energy (such as heat contained in air and soil and water) which cannot be directly utilized into high-level heat energy which can be utilized, thus achieving the purpose of saving some of the high-level energy (such as coal, gas, oil, electric energy, etc.). As a low-level heat source heat pump, air is inexhaustible and available everywhere for free. Moreover, the installation and use of air-source heat pumps are quite convenient. At present, products using ASHP include domestic heat pump AC, commercial unit heat pump AC unit and heat pump cold and hot water units, etc.

The ASHP has the following characteristics. (1) The ASHP system combines heat and cold sources and does not require a special chiller room or boiler room. The unit can be

placed on the roof or ground at will, without occupying the effective usage area of the building, so the construction and installation are very simple. (2) The ASHP system has no cooling water system, no cooling water consumption and no power consumption for a cooling water system. In addition, many infections caused by cooling water pollution have been reported. Considering safety and sanitation, ASHP has obvious advantages. (3) The ASHP system is safe, reliable and environment-friendly because it does not need boilers, a corresponding boiler fuel supply system, dust removal system and a flue gas exhaust system. (4) The modular design is adopted for the ASHP cooling (hot) water unit. No standby unit is required. During operation, the computer controls automatically and regulates the operation state of the unit to adapt the output power to the working environment. (5) The performance of ASHPs varies with outdoor climate change. (6) In places with low outdoor air temperature, an auxiliary heater is required due to insufficient heat supply from the heat pump in winter.

The ASHP water heater is a hot pump water heater which uses air as a low temperature heat source to produce domestic hot water. It consists of an ASHP circulation system and water storage tank. The ASHP water heater is a device which transfers heat from air to water to produce hot water by consuming part of the electric energy. Under the condition of evaporation temperature of 0 °C, the COP value of the $CO_2$ heat pump hot water system can reach 4.3 by heating water from 9 °C to 60 °C. With ambient air as the heat source, the average operating COP value for the whole year can reach 4.0, which can save 75% of the energy compared with conventional electric heating or coal-fired boilers. Figure 4 illustrates the application research status of machine vision technology in Exchangers.

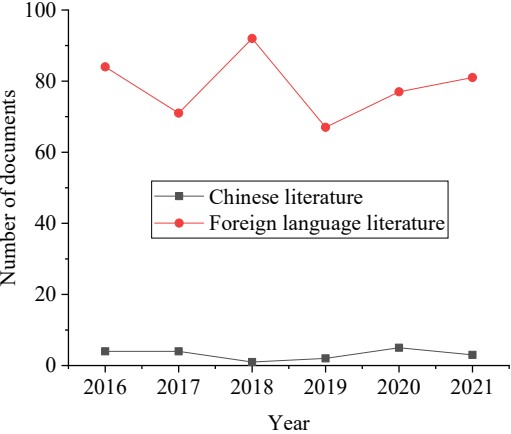

**Figure 4.** Research status of machine vision in Exchanger application.

As shown in Figure 4, the scope of machine vision technology applied in Exchanger research is already very wide. The number of foreign research papers has risen to 92 per year and the number of Chinese research papers is to 5. Therefore, abundant research is needed for future development to support the optimization of its application.

### 3.2. Application of MV in Defrosting System

MV technology is an important field of AI. It uses visual sensors instead of human eyes to obtain the object image in a Three-Dimensional (3D) space and to extract, process and analyze the target information from the digital image by computer. The introduction of MV technology is relatively late in China. Not until the 1990s were MV technology companies established. Now, MV is becoming popular and seeing applications such as Automatic License Plate Recognition (ALPE) and object surface defect detection. Nevertheless, existing MV approaches have presented low accuracy and single functionality. MV-based recognition technology has been relatively mature in developed nations such as European countries and the United States. Their MV-enabled industrial applications have been fully-fledged. Under such a backdrop, the Chinese government has issued relevant

favorable policies to promote the domestic development of MV technologies. For example, in automobile manufacturing, especially in the New Energy Vehicle (NEV) production and renewable energy fields, MV techniques are being applied more broadly. The application fields of MV technology are encapsulated in Figure 5:

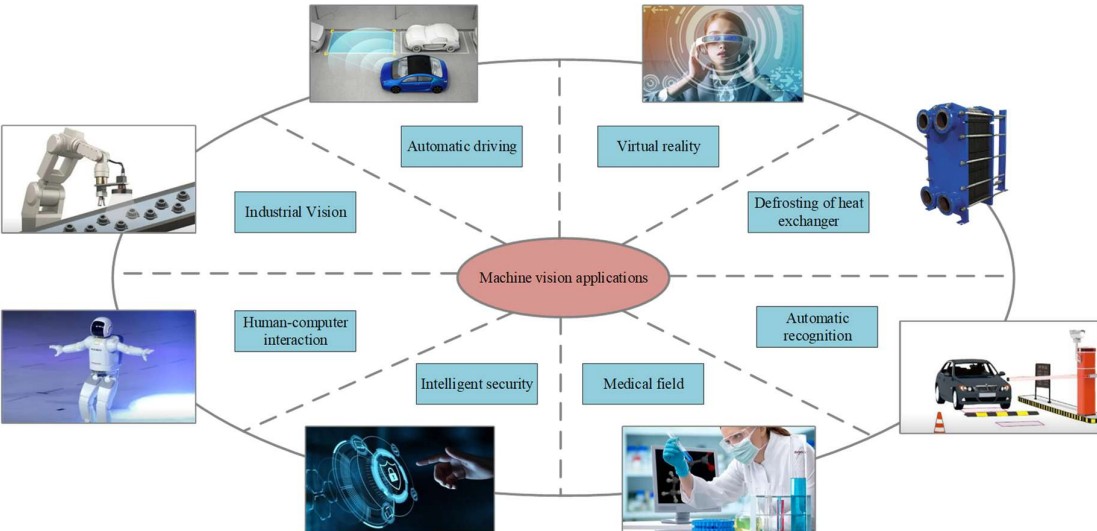

**Figure 5.** Application fields of MV.

He et al. (2021) et al., applied the PEC principle to the ASHP defrosting system. The frost layer's light transmittance varied with its thicknesses. The PEC could output the corresponding voltage signal through the induction of the frost layer. The experiment noticed that the proposed technology accurately measured the frost layer growth. Suppose air temperature, humidity and wind speed were 17.7–27.2 °C, 21.6–64.9% and 0.5–4.5 m/s and the outdoor Exchanger's surface temperature was −10.5–6.5 °C. In that case, the frost layer thickness had a linear relationship with the photoelectric sensor's output voltage. The error between the corrected equation and the experimental results was less than 5%. The result proved that the proposed method was feasible as the defrosting criterion and had more robust adaptability to the harsh environment than other direct measurement methods. Finally, it was pointed out that the installation position of the photoelectric sensor had a significant influence on the frost layer monitoring state [42].

The Performance Requirements and Test Methods of the Automobile Windshield Defrosting and Defogging System drafted by Changchun Automobile Research Institute in 1994 and the Performance Requirements and Test Methods of the Automobile Windshield Defrosting and Defogging System describe the Executive Standards of automobile windshield defrosting and defogging detection system in China. An essential part of the windshield defrosting detection system is detection of the windshield defrosting area.

In the actual detection of window defrosting process, most of the operating procedures in the detection process rely on manual operation to complete. First, the inspector draws the melted profile of the frost with a straight marking on the windscreen of the car. This process also records it several times in the same time interval to obtain the profile of the melted area at different times, leaving a process map of the melting of the frost on the windscreen of the car. The recording ends when all the frost on the window has melted. The frost melting process depicted on the window is then rubbed in a 1:1 scale and the resulting plan is captured with a camera and imported into the plan drawing software. When calculating an irregular defrosting area in a digital image by computer software, its calculation speed is significantly improved compared with actual measurement and the accuracy of the results is high. Finally, the inspector calculates the area in the digital image and enlarges it proportionally to obtain the defrost area on the windscreen of the

vehicle. Figure 6 reveals the research status of the application of machine vision technology in defrosting.

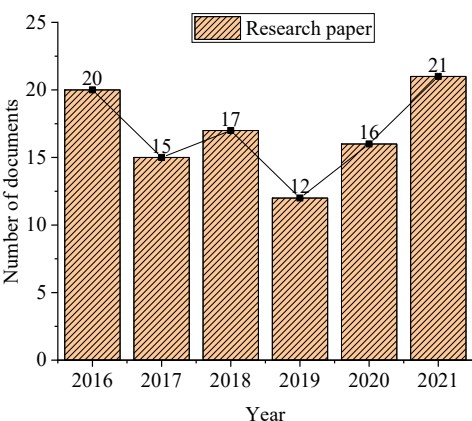

**Figure 6.** Research status of the application of machine vision technology in defrosting.

As shown in Figure 6, there are not many research projects on the application of machine vision technology in defrosting at present (maximum number 21). The current social research on the application of machine vision technology in defrosting is not mature and further research is needed to provide technical reference for its development.

### 3.3. Challenges and Prospects of Exchanger Defrosting Control

#### 3.3.1. Challenges

Today, China proposes a "double carbon" goal and pays increasing international attention to environmental problems. Accordingly, the market of ASHP will be expanded and its application potential will be further tapped. However, the application and promotion of ASHP AC still face many challenges from policy, supervision, economy, technology and public acceptance levels, as expanded in Figure 7:

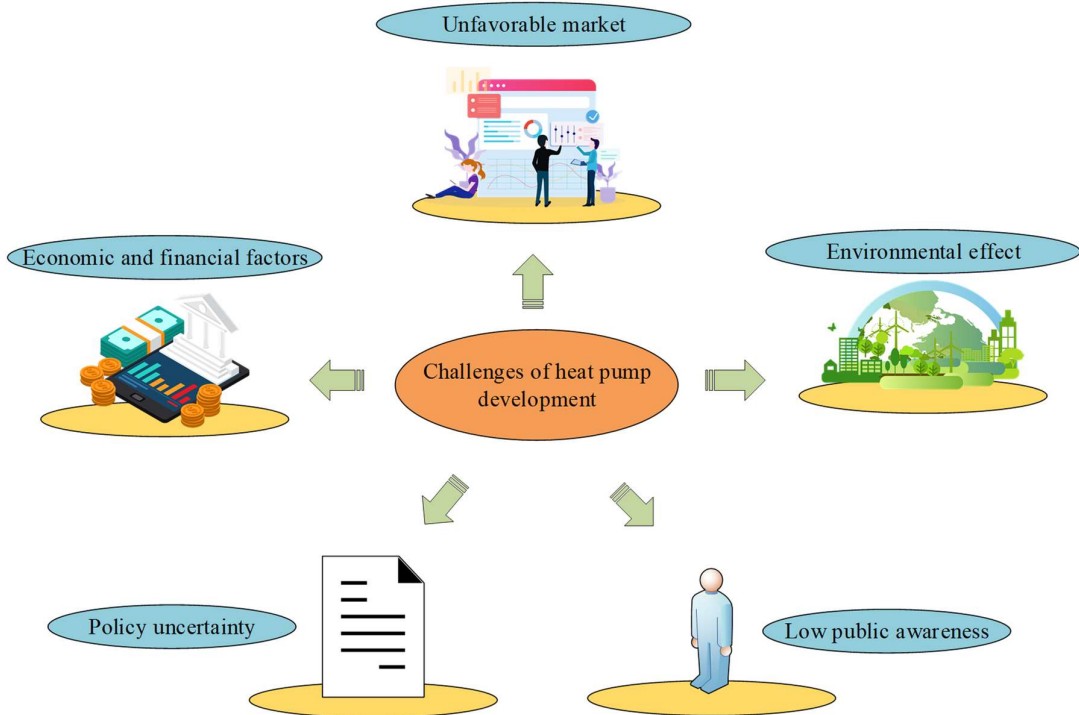

**Figure 7.** Challenges faced by the development of ASHP AC.

With innovations in technology, some defrosting control methods are based on intelligent algorithms, such as NN, fuzzy intelligent control and self-correction. In particular, the NN method sets the input variables based on a specific algorithm. It establishes the NN structure to predict the defrosting interval, defrosting time and frosting amount. Ding et al. (2021) factored in NN's fan running time and outdoor air physical parameters. They established a multilayer feedforward NN model trained by the Backpropagation (BP) algorithm to predict the frost amount and train and test the structure of the model. It was found that the experimental results were very close to the measured values [43]. Liu et al. (2018) predicted the nominal heating loss coefficient of ASHP using a generalized NN. The model correlation coefficient was greater than 0.9, the expected deviation percentage was less than 6.5% and the model's general ability and learning and training effect were excellent [44].

The fuzzy intelligent control method can defrost according to appropriate operating conditions but needs to formulate fuzzy control rules and fuzzy reasoning blocks according to extensive experimental data. Self-correction method has strong reliability and scientific basis but needs sufficient and reliable sample data to form correction rules and defrosting control rules.

### 3.3.2. Prospects

(1) Direct observation can measure the frost layer thickness on the Exchanger surface and improve accuracy. Nevertheless, it is affected by cost and operation space and it is not easy to obtain a better practical application effect for ASHP units. In contrast, indirect monitoring is simple, has few additional requirements and is widely used in ASHP monitoring. The MV-based Exchanger defrosting control method needs large volumes of training samples for formulating defrosting control logic. Surging AI and Big Data Technology (BDT) provide a more intelligent approach to researching the MV-based Exchanger surface defrosting and condensation control.

(2) Delaying or suppressing Exchanger surface frosting from the source can reduce energy consumption, increase RUR and promote the development of ASHP units. Frosting can be suppressed by setting frost coating on Exchanger surfaces or changing its structures and shapes. Reasonable defrosting approaches ensure a comfortable indoor temperature, minimize energy wastage, maximize unit stability and shorten defrosting time.

### 4. Conclusions

With the development of society and people's living standards, demand for air-conditioning products is increasing. Air-conditioning refrigeration and heating are required in most areas of China to ensure comfort in the working and living environment. ASHP is extensively used in energy reform of the "coal to electricity" project because of its low cost, convenient installation and use, high efficiency and absence of environmental problems. This study analyzes the formation and growth mechanism of surface defrosting on heat exchangers, summarizes the current research status of machine vision technology in defrosting and dew problems and discusses the follow-up research direction. Finally, the application research status of machine vision technology in frost dew phenomenon in current society is analyzed through literature retrieval, providing important reference for the application and research development of future machine vision technology in frost dew phenomenon. Here, machine vision technology is similar to manual observation in observing the characteristics of frost dew phenomenon. At the same time, it avoids the shortcomings of low observation frequency and lack of objectivity in manual observation. From the point of view of scientific research, there is still a large space for automatic observation of frost-dew phenomenon. Although this work provides comprehensive research content, there is no specific research concept on how to carry out research and development. Therefore, future research will strengthen the specific application research of machine vision in frost-dew phenomenon to promote further the development of this technology.

**Author Contributions:** Conceptualization, B.Y.; methodology, B.Y., X.Z.; investigation, X.Z., M.L. and Z.L.; resources, Z.L.; data curation, X.Z. and M.L.; writing—original draft preparation, B.Y. and Z.L; writing—review and editing, Z.L.; supervision, B.Y.; project administration, Z.L. All authors have read and agreed to the published version of the manuscript.

**Funding:** This research received the National Natural Science Foundation of China (No. 52278119).

**Institutional Review Board Statement:** Not applicable.

**Informed Consent Statement:** Not applicable.

**Data Availability Statement:** Data available on request due to restrictions e.g., privacy or ethical.

**Conflicts of Interest:** The authors declare no conflict of interest.

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
