# Peer review of "Review on the Application of Machine Vision in Defrosting and Decondensation on the Surface of Heat Exchanger"

_sustainability, doi:10.3390/su141811606_

Round 1

Reviewer 1 Report

This work reviews the MV technology in defrosting and dew decondensation. The following comments should be addressed before acceptance.

1.     Since this work focuses on the defrosting of ASHP, more information of ASHP should be given, such as application area, climate factor, economical factor, etc.

2.     In introduction section, the authors mentioned Thermal defrosting and mechanical defrosting are two widely used defrosting techniques.” Therefore, the authors should review the research status of these two defrosting techniques rather than internal and external methods.

3.     Section 2.2.1 and section 2.2.2 have same titles.

4.     The authors should cite more data for demonstrating the improvement of the Exchanger’s performance via frost suppressing. Also, more data should be cited for demonstrating the advantage of applying MV in defrosting.

Author Response

Reviewer1

Comments and Suggestions for Authors

This work reviews the MV technology in defrosting and dew decondensation. The following comments should be addressed before acceptance.

  1.     Since this work focuses on the defrosting of ASHP, more information of ASHP should be given, such as application area, climate factor, economical factor, etc.

Reply: Thank you for your suggestion. We have added a discussion of ASHP in Section 3.1.2 of the text as suggested.

  1.     In introduction section, the authors mentioned “Thermal defrosting and mechanical defrosting are two widely used defrosting techniques.” Therefore, the authors should review the research status of these two defrosting techniques rather than internal and external methods.

Reply: Thanks for your comment. We have revised the Introduction section with a full discussion of the research direction of this paper, reducing the need for research on the sentence "thermal defrosting and mechanical defrosting are two widely used defrosting techniques" influence of content.

  1.     Section 2.2.1 and section 2.2.2 have same titles.

Reply: Thank you for your reminder. We have modified the headings of Section 2.2.1 and Section 2.2.2, thus improving the plausibility of the article.

  1.     The authors should cite more data for demonstrating the improvement of the Exchanger’s performance via frost suppressing. Also, more data should be cited for demonstrating the advantage of applying MV in defrosting.

Reply: Thanks for your guidance. We have added Figure 4 and Figure 6 to the text as suggested, in which the application status of the current frost and dew research on machine vision technology is provided, thereby providing more data support for this research.

Reviewer 2 Report

Review manuscript for Sustainability 

Recommendation

Accept after minor revision 

Comments:

Manuscript ID: sustainability-1897681

Title: Review on the Application of Machine Vision in Defrosting and Decondensation on the Surface of Heat Exchanger. 

The present manuscript first analyzes the formation and growth mechanism of Exchanger surface frosting and condensation. It then summarizes the current research status of Machine Vision (MV) technology in defrosting and decondensation. Further, it prospects the follow-up research direction.

Overview and general recommendation:

1. Overall a good review manuscript, as it explains in detail the process and issues, the evolution the technology and a good outlook on the future.

2. In the introduction they only have a reference, they should rely more on previous works.

3. In section 2 and other sections of background groups, it is recommended to make tables with the different studies, including the researchers, year, characteristics, contributions, among others.

4. In line 134, it is necessary to define before using the term DLA.

5. In line 179, why start with (2)?

6. At the end of line 206, there is one ¨e¨ too many.

7. The conclusions should be more based or argued in the analysis of what was presented. Much of the text is about what was done and not what is concluded.

Author Response

Reviewer2

Comments and Suggestions for Authors

Review manuscript for Sustainability

Recommendation

Accept after minor revision

Comments:

Manuscript ID: sustainability-1897681

Title: Review on the Application of Machine Vision in Defrosting and Decondensation on the Surface of Heat Exchanger.

The present manuscript first analyzes the formation and growth mechanism of Exchanger surface frosting and condensation. It then summarizes the current research status of Machine Vision (MV) technology in defrosting and decondensation. Further, it prospects the follow-up research direction.

Overview and general recommendation:

  1. Overall a good review manuscript, as it explains in detail the process and issues, the evolution the technology and a good outlook on the future.

Reply: Your affirmation is an important support for the publication of this article. We would like to express our deep gratitude to your guidance.

  1. In the introduction they only have a reference, they should rely more on previous works.

Reply: This is a good reminder. We have revised the second paragraph of the Introduction section to provide a discussion of the relevant research as required.

  1. In section 2 and other sections of background groups, it is recommended to make tables with the different studies, including the researchers, year, characteristics, contributions, among others.

Reply: Thank you for your suggestion. This article has already provided specific relevant research data in Section 3, and therefore does not provide information on relevant studies in Section 2.

  1. In line 134, it is necessary to define before using the term DLA.

Reply: Thank you for your reminder. The DLA was already defined earlier in this article when it refers to the DLA on line 134.

  1. In line 179, why start with (2)?

Reply: Thanks for your comment. We have removed (2) in line 179.

  1. At the end of line 206, there is one ¨e¨ too many.

Reply: Thanks for your opinion. We have deleted the extra “e” in line 206 according to your comments.

  1. The conclusions should be more based or argued in the analysis of what was presented. Much of the text is about what was done and not what is concluded.

Reply: Thanks for your careful reading. We have revised the conclusion of the article and highlights the research content of the article as suggested, thereby enhancing the value of the conclusion section.

Round 2

Reviewer 1 Report

The comments have been addressed.